# The Association of Placental Abruption and Pediatric Neurological Outcome: A Systematic Review and Meta-Analysis

**DOI:** 10.3390/jcm12010205

**Published:** 2022-12-27

**Authors:** Irina Oltean, Ajay Rajaram, Ken Tang, James MacPherson, Tadiwanashe Hondonga, Aanchal Rishi, Regan Toltesi, Rachel Gowans, Ashkan Jahangirnia, Youssef Nasr, Sarah L. Lawrence, Dina El Demellawy

**Affiliations:** 1Department of Surgery & Pathology, Children’s Hospital of Eastern Ontario, Ottawa, ON K1H 8L1, Canada; ioltean@cheo.on.ca (I.O.); ktang@cheo.on.ca (K.T.); 2Department of Pathology, McGill University, Montreal, QC H4A 3J1, Canada; ajay.rajaram@mail.mcgill.ca; 3Department of Pathology and Laboratory Medicine, University of Ottawa, Ottawa, ON K1H 8M5, Canada; jmacpherson@eorla.ca (J.M.); yousseflotfynasr@gmail.com (Y.N.); 4Department of Health Sciences, Carleton University, Ottawa, ON K1S 5B6, Canada; tadihondonga@cmail.carleton.ca; 5Schulich School of Medicine & Dentistry, Western University, London, ON N6A 5C1, Canada; arishi3@uwo.ca; 6Faculty of Science, Engineering and Architecture, Laurentian University, Sudbury, ON P3E 2C6, Canada; rtoltesi@laurentian.ca; 7Faculty of Health Sciences, University of Ottawa, Ottawa, ON K1N 6N5, Canada; rgowa085@uottawa.ca; 8Faculty of Medicine, University of Ottawa, Ottawa, ON K1H 8M5, Canada; ajaha039@uottawa.ca (A.J.); sllawrence@cheo.on.ca (S.L.L.); 9Division of Neonatology, Children’s Hospital of Eastern Ontario, Ottawa, ON K1H 8L1, Canada; 10Department of Pathology and Laboratory Medicine, Children’s Hospital of Eastern Ontario, Ottawa, ON K1H 8L1, Canada

**Keywords:** infant, child, abruptio placentae, pregnant women, cerebral palsy, morbidity, delivery, obstetric hemorrhage

## Abstract

Placental histopathology provides insights, or “snapshots”, into relevant antenatal factors that could elevate the risk of perinatal brain injury. We present a systematic review and meta-analysis comparing frequencies of adverse neurological outcomes in infants born to women with placental abruption versus without abruption. Records were sourced from MEDLINE, Embase, and the CENTRAL Trials Registry from 1946 to December 2019. Studies followed the PRISMA guidelines and compared frequencies of neurodevelopmental morbidities in infants born to pregnant women with placental abruption (exposure) versus women without placental abruption (comparator). The primary endpoint was cerebral palsy. Periventricular and intraventricular (both severe and any grades of IVH) and any histopathological neuronal damage were the secondary endpoints. Study methodologic quality was assessed by the Ottawa–Newcastle scale. Estimated odds ratios (OR) and hazards ratio (HR) were derived according to study design. Data were meta-analyzed using a random effects model expressed as pooled effect sizes and 95% confidence intervals. We included eight observational studies in the review, including 1245 infants born to women with placental abruption. Results of the random effects meta-analysis show that the odds of infants born to pregnant women with placental abruption who experience cerebral palsy is higher than in infants born to pregnant women without placental abruption (OR 5.71 95% CI (1.17, 27.91); *I*^2^ = 84.0%). There is no statistical difference in the odds of infants born to pregnant women with placental abruption who experience severe IVH (grade 3+) (OR 1.20 95% CI (0.46, 3.11); *I*^2^ = 35.8%) and any grade of IVH (OR 1.20 95% CI (0.62, 2.32); *I*^2^ = 32.3%) vs. women without placental abruption. There is no statistically significant difference in the odds of infants born to pregnant women with placental abruption who experience PVL vs. pregnant women without placental abruption (OR 6.51 95% CI (0.94, 45.16); *I*^2^ = 0.0%). Despite our meta-analysis suggesting increased odds of cerebral palsy in infants born to pregnant women with placental abruption versus without abruption, this finding should be interpreted cautiously, given high heterogeneity and overall poor quality of the included studies.

## 1. Introduction

Placental abruption can lead to bleeding at the decidual placental interface. The rate of placental abruption in several European countries has declined, while the trend in North America is increasing [1]. Placental abruption remains a critical concern for maternal and perinatal mortality, as well as morbidity. It can be associated with trauma and drug use. Maternal complications due to abruption vary based on its severity [2] and can include outcomes such as myocardial infarction, cardiomyopathy, heart failure, cerebrovascular disorders and even death. With respect to perinatal outcomes, lower gestational age can predict perinatal severity in pregnant women with abruption [3]. Pregnant women with abruption are more likely to have preterm births (i.e., live births earlier than 37 weeks of gestation) than women without abruption [4]. Intrauterine growth retardation (IUGR), congenital abnormalities [5], and complications of prematurity are additional neonatal consequences of placental abruption.

Placental histopathology provides insights, or “snapshots”, into relevant antenatal factors that could elevate the risk of perinatal brain injury [6]. In general, perinatal mortality and maternal complications occur in situations of severe placental abruption and intrauterine death [7]. In severe placental abruption cases, obstetricians must decide whether to intervene early and perform a caesarean delivery, which poses risks to hemodynamically unstable mothers, or delay fetal delivery, which increases the risk of fetal mortality and neurological morbidity. Evidence demonstrates that when fetal heart patterns are normal, delaying delivery when the maternal status is stable can be appropriate [8,9,10].

Several studies on long-term consequences of abruption have focused on non-neurological infant consequences [11,12,13,14] or maternal cardiac outcomes [15,16,17,18] with less focus on neonatal neurological impairment. There is a relationship between placental abruption and the possibility for fetal neurologic impairment. Greater than 50% of placentas from medicolegal cases with neurologic impairment had fetal thrombotic vasculopathy [6], which is a known etiology of placental abruption. Whitehead et al. [19] reported a significant association between abruption and childhood epilepsy (RR 4.6, 95% CI 3–6.9) in large, population-based cohorts, and this relationship holds true across the literature [20,21,22,23,24]. A small number of studies have investigated brain-related injuries, such as neonatal encephalopathy (NE)/hypoxic ischemic encephalopathy (HIE) [25,26,27]. No association was found between placental abruption and intraventricular and periventricular hemorrhage in babies born preterm [28,29,30]. Despite this, long-term risks such as cerebral palsy [21,31,32,33,34] and cognitive deficits [35] were elevated in surviving neonates. The association between abruption and cerebral palsy is, however, inconclusive. Additional studies show no difference in risk between neonates in pregnancies with and without abruption, although study populations differ (i.e., women without chronic hypertension vs. spastic cerebral palsy only vs. deliveries between 22 and 26 weeks) [30,36,37].

Although one systematic review has examined neonatal neurological outcomes in a setting of placental abruption [38], the number of included studies focusing on neonatal outcomes in relation to obstetric risks were minimal. In order to supplement this current body of knowledge, we performed a systematic investigation into the unique interplay between placental abruption and long-term neurological infant consequences.

Based on our search criteria, there is no current meta-analysis performed in 2021 that examines the neurological consequences of placental abruption. Therefore, the objective of this present study is to statistically review the primary literature examining infant neurological consequences among pregnant women with placental abruption versus without placental abruption.

## 2. Materials and Methods

This review followed the Cochrane Methodology to identify and select the studies [39] and the Preferred Reporting Items for Systematic Reviews and Meta-Analyses (PRISMA) to guide the reporting of this systematic review [40].

### 2.1. Eligibility Criteria, Information Sources, Search Strategy

A systematic search for relevant studies was performed between 1946 and December 2019, using the following databases: MEDLINE including Epub Ahead of Print, In-Process and Other Non-Indexed Citations (1946–December 2019) and Embase (1947–December 2019) and the CENTRAL Trials Registry of the Cochrane Collaboration (December 2019 Issue) using the Ovid interface. The rationale for including studies from 1946 to 2019 was to ensure all relevant data were captured, although we recognize that more recent studies are ideal given their greater clinical impact.

Searches were limited to English or French and full journal articles. Searches were designed, conducted, and pre-specified by a librarian experienced in systematic reviews, using a method designed to optimize term selection [41]. Prior to screening and data analysis, this review was registered in PROSPERO [42] on December 16, 2019 (registration number CRD42020161913).

After duplicate records were removed online, records retrieved by the electronic search were downloaded and imported into a reference manager, and then uploaded to a systematic review software InsightScope (InsightScope.ca) for title and abstract screening and full text review. Five reviewers (I.O., A.R., T.H., A.R., R.T.) screened at the title/abstract level and six reviewers (I.O., A.R., T.H., A.R., R.T., R.G.) at the full text review stages. Citations were excluded if at least two reviewers agreed to exclude.

### 2.2. Study Selection

Case-control and cohort studies examining our outcomes of interest, specifically infant hypoxic ischemic encephalopathy (HIE), periventricular leukomalacia (PVL), intraventricular hemorrhage (IVH), presence or absence of brain damage, and cerebral palsy among infants (<2 years) born to pregnant women with any stage of placental abruption in the first, second or third trimester versus without abruption, were included. Data were extracted by two reviewers (I.O. and A.R.) and verified by the opposite reviewer with no need for third-party consultation. Studies were excluded if our outcomes of interest were not mentioned (n = 90), or if there was ineligible exposure (i.e., absence of placental abruption) (n = 2), or other (i.e., cerebral palsy and death frequencies were combined and could not be teased separately) (n = 1).

### 2.3. Data Extraction

Two authors (I.O. and A.R.) extracted data using a pre-designed and piloted data abstraction sheet in REDCap (Research Electronic Data Capture) [43]. Briefly, the extracted information included: frequency of pregnant women with or without placental abruption, demographics such as maternal age, parity, smoking, gestational age, intraventricular hemorrhage (IVH), presence or absence of periventricular leukomalacia (PVL), cerebral palsy (CP), brain damage (histopathological evidence of neuronal damage in the brain), hypoxic ischemic encephalopathy (HIE), and neonatal mortality. Of note, we included all stages of IVH (not solely severe III and IV forms but I and II as well) to minimize selection bias and to capture cases of mild IVH and placental abruption as described in our included studies.

Our exposure variable was a diagnosis of any stage of placental abruption in the first, second or third trimester of pregnancy (based on gestational age in weeks). Four of the included studies [44,45,46,47] assessed mild, moderate, or severe abruption in the third trimester using the classification systems from Page [48] or Knab [49]. One study focused on grade 1 (external bleeding and mild uterine pain) and grade 2 (grade 1 symptoms, concealed hemorrhage, uterine tetany, and/or fetal distress but no maternal shock) placental abruptions occurring during the third trimester [50]. Another study examined any case of placental abruption among stillborn infants born at gestation of 24 or more weeks [51]. Two studies used clinical definition for placental abruption, in particular, the premature detachment of an implanted placenta from the uterine wall before delivery of the fetus, in the third trimester of pregnancy [24]. One study included any case of placental abruption, defined as the presence of retroplacental hematoma and clinical presentations (any, or the combination of genital bleeding, abdominal pain, pregnancy-induced hypertension, premature labor, premature rupture of membranes, intrauterine fetal death, or non-reassuring fetal heart rate pattern) in the second trimester of pregnancy [30].

The primary endpoint was cerebral palsy (CP). The secondary endpoints included IVH (both severe and any grades of IVH), PVL, and any histopathological neuronal damage (as defined by the study authors). Among our included studies, a child of at least 2 years at diagnosis, validated by physical findings and confirmed by a pediatrician, was considered a confirmed case of cerebral palsy [30,47]. A diagnosis of cerebral palsy could also have been made using the neurodevelopmental evaluation of the Bayley Scales of Infant Development at discharge and after 3, 6, 12 and 24 months. One study examined hospitalizations and further defined the diagnosis of cerebral palsy as the presence of unspecified infantile cerebral palsy, paraplegia, diplegia of upper limbs, or unspecified paralysis. All studies examining IVH used neonatal cranial ultrasound for diagnosis within 24 to 48 h on admission or at birth and weekly until discharge from the hospital [30] or cranial ultrasound at 3, 6, 12 or 24 months after birth (corrected gestational age) [45]. HIE was confirmed using CT or MRI and by clinical exam, and a severe diagnosis of HIE was made if the infant experienced frequent seizures, apnea, flaccid weakness, or coma [46]. Evidence of established brain damage included immunohistochemical staining of post-autopsy brain specimens to visualize antibodies to glial fibrillary acidic protein for astrocytes and to CD68 and major histocompatibility complex II for microglia [51]. We selected the above neurological outcomes because their consideration specifically appears to be lacking in the obstetric literature with respect to pregnant women with placental abruption.

### 2.4. Assessment of Risk of Bias

Three factors were considered to score the quality of included studies: (1) Selection, including representativeness of the exposed cohort, selection of the non-exposed cohort, ascertainment of exposure, and demonstration that at the start of the study, the outcome of interest was not present. (2) Comparability, assessed on the basis of study design and analysis, and whether any confounding variables (not necessarily all) were adjusted for in a regression model (including gestational age, social class, or duration of mother’s education, in addition to other confounders such as previous or current abuse of substances and presence of medical comorbidities). Of note, we did not include birthweight as a potential confounder, as it may be a causal intermediate and thus contribute to collider bias [52]. Gestational age and birth weight are highly collinear, which makes it difficult to distinguish their individual effects on neurological outcome [53]. (3) Outcome, based on the follow-up period for when neurological outcome can be detected (minimum 24 months), cohort retention, and assessment of neurological outcome through independent blind assessment to the diagnosis of placental abruption. In other words, the pediatrician or neurologist who assesses the neurological outcome of the infant or child cannot be aware of the diagnosis of placental abruption in the mother. Conversely, the pathologist should not be aware of the clinical history of the case when making a diagnosis of neurological impairment.

We rated the quality of the studies (good, fair, and poor) by awarding stars in each domain following the guidelines of the Newcastle–Ottawa Scale [54]. A “good” quality score required 3 or 4 stars in selection, 1 or 2 stars in comparability, and 2 or 3 stars in outcomes. A “fair” quality score required 2 stars in selection, 1 or 2 stars in comparability, and 2 or 3 stars in outcomes. A “poor” quality score reflected 0 or 1 star(s) in selection, or 0 stars in comparability, or 0 or 1 star(s) in outcomes [55].

### 2.5. Data Synthesis

All data analyses were performed using the R statistical programming language (Version 4.0.3) [56]. Dichotomized variables were expressed as numbers and percentages. Data was meta-analyzed using a random effects model with R package “meta” [57]. Odds ratios (OR) were derived from case control studies. The estimated hazards ratio (HR) was used for one study whose HR could not be converted to OR [24]. When available in individual studies, control groups with placental previa were included. Statistical heterogeneity was determined using *I*^2^ tests. *I*^2^ is the proportion of total variation observed between studies attributable to differences between studies rather than sampling errors. We considered high heterogeneity if *I*^2^ > 75%.

## 3. Results

### 3.1. Study Selection

After initial screening, 101 articles were retained for full-text screening, of which 8 records met all the study criteria (i.e., captured at least one of our outcomes of interest) (Figure 1).

### 3.2. Study Characteristics

Six of the eight studies were retrospective case-control studies, with sample sizes ranging from 53 to 217,910 (Table 1). There were 1245 infants born to women with placental abruption versus 217,608 without placental abruption. The mean maternal age of the participants with placental abruption was 29.4 ± 6.1 years vs. 28.2 ± 5.7 years in those without abruption. The mean gestational age of infants born to pregnant women with abruption was 35.4 ± 4.4 vs. 39.1 ± 1.8 weeks in those without abruption. Table 2 and Table 3 demonstrate the maternal and neonatal characteristics of the study population (depending on data availability).

### 3.3. Risk of Bias of Included Studies

There were four poor, one fair, and three good-quality studies. Three studies adjusted for confounders in a regression model, awarding them a good evaluation [24,44,45]. Only two of the eight studies had a follow-up for at least 24 months of corrected gestational age [24,45]. Case-control studies should have clearly detailed when the neonatal examination occurred for adequate follow-up of a neurological diagnosis [44]. Pariente et al. [24] had completed follow-up of all the study participants, but in Spinillo et al. [45], the follow-up rate was less than 95% and with no description of those lost to follow-up. The presence of placental abruption in all studies was ascertained by referencing medical records or computer databases. In three studies, histopathological examination of the placenta was also used to confirm the presence of abruption [24,45,58], while only clinical confirmation was used in the others. None of the studies demonstrated any assessments to exclude the presence of study outcomes prior to commencement of the study. Blinding during outcome assessment was performed only in two studies, where pediatric neurologists were blinded during neurodevelopmental assessment of children with cerebral palsy [30] or where pathologists performing autopsies were blinded to clinical histories [51] (Table 4).

### 3.4. Synthesis of Results

Of the eight studies that included one of our outcomes of interest (i.e., cerebral palsy, IVH or PVL), only five were suitable for meta-analysis [24,30,45,50,58]. At least two of the five studies directly compared the risk of one of our outcomes of interest. Spinillo et al., 1993 [44], could not be added to avoid the possibility of patient overlap, and only one study measured brain damage at autopsy [51]. Four studies specifically reported on the association of abruption with cerebral palsy [24,30,45,58]. Since we had fewer than 10 studies in this meta-analysis, we did not perform a meta-regression based on the Cochrane handbook [59] to examine the effect of gestational age and placental abruption on cerebral palsy. Therefore, we could not examine the impact of decreasing gestational age on birth asphyxia and consequently the risk of cerebral palsy [60,61].

Results of the random effects meta-analysis show that the odds of infants born to pregnant women with placental abruption who experience cerebral palsy are higher than in infants born to pregnant women without placental abruption (OR 5.71 95% CI (1.17, 27.91); *I*^2^ = 84.0%). However, the confidence interval is very wide and imprecise (Figure 2). Pooled effect sizes were used to approximate the OR rather than the HR, since Pariente et al. had <50% weight in this meta-analysis.

Due to the confidence interval including one, there is no statistical difference in the odds of infants born to pregnant women with placental abruption who experience severe IVH (grade 3+) vs. women without placental abruption (OR 1.20 95% CI (0.46, 3.11); *I*^2^ = 35.8%) (Figure 3). Likewise, there is no statistical difference in the odds of infants born to pregnant women with placental abruption who experience any grade of IVH than in infants born to pregnant women without placental abruption (OR 1.20 95% CI (0.54, 2.68); *I*^2^ = 32.3%) (Figure 4).

Only two studies reported on the association of abruption and PVL. Resulting from very wide, imprecise confidence intervals including one, there is no statistically significant difference in the odds of infants born to pregnant women with placental abruption who experience PVL vs. pregnant women without placental abruption (OR 6.51 95% CI (0.94, 45.16); *I*^2^ = 0.0%) (Figure 5).

Excluded from the meta-analysis were neonatal mortality [30,58] and HIE [46] due to an insufficient number of studies. In Furukawa et al., there were no differences in the incidence of neonatal death in women with placental abruption versus without. Matsuda et al. determined that neonatal mortality was more frequent in cases of placental abruption (11/42, 26.2%) than in placenta previa (2/72, 2.8%) and preterm labor (1/120, 0.8%), although the difference was predominantly due to cerebral palsy. HIE was only reported in one study [46], with higher incidences of moderate and severe HIE in the placental cases than in healthy term newborns.

## 4. Discussion

### 4.1. Principal Findings

Our random effects meta-analysis showed statistically significant increased odds of cerebral palsy (CP) in infants born to pregnant women with abruption versus those born to mothers without abruption, which align with prior literature. Of note, we reported very high heterogeneity for this finding, likely due to variations in the measurement of placental abruption diagnosis (exposure) and cerebral palsy. In light of these variations, this particular finding should be interpreted with caution. Our meta-analysis also shows no statistical difference in the increased odds of any grade and severe IVH, respectively, in infants born to pregnant women with placental abruption. We noted no difference in the odds of PVL in infants born to pregnant women with placental abruption. However, this result was not statistically significant and imprecise.

### 4.2. Comparison with Existing Literature

In line with our meta-analysis, other studies have shown a similar increase in CP [38,62]. After adjusting for confounding factors, Spinillo et al. [45] found that infants from pregnancies complicated by placental abruption were 3.9 times more likely than controls to have neonatal death or cerebral palsy. This finding was echoed in the long-term follow-up study conducted by Pariente et al. [24]. Infants born to mothers with placental abruption showed higher rates of cerebral palsy (0.73 vs. 0.10 per 1000 person-years) than the control group without abruption. Of all the selected studies, only Furukawa et al. [30] showed no significant difference in the incidence of cerebral palsy between the abruption and control groups.

Several studies in the literature have shown a direct relationship between placental abruption and cerebral palsy [7,21]. The precipitating and pathologic factor contributing to the development of cerebral palsy is fetal acidemia, resulting from impaired gas exchange and endothelial dysfunction, as reflected by abnormal umbilical pH/cord gases. Placental abruption is associated with acidosis, impaired gas exchange and endothelial dysfunction [63]. Since neuronal tissue is sensitive to metabolic disturbances [64], placental abruption could lead to hypoxia, hypercapnia, and in turn, CP. Moreover, studies have shown lower umbilical cord blood pH after placental abruption in the CP group than in the non-CP group [22]. In cases with umbilical pH < 7.0, neonates are at a higher risk of HIE due to acidemia, which in turn, elevates their risk of CP [65,66]. Acidemia (pH ≤ 7.0) [67] occurred in 114 of 168 (69.2%) cases of severe CP, and a majority of these were also pregnancies complicated by placental abruption [68]. Another report documented that more than half of all CP cases showed umbilical cord blood pH < 7.0, further supporting the link between fetal acidemia, abruption, and CP [69].

Risk factors for abruption include maternal habits such as alcohol consumption and smoking during pregnancy, as well as multiparity and hypertensive disorders in pregnancy [70,71,72]. In particular, alcohol consumption (OR 3.38, 95% CI (2.01–5.68)), smoking during pregnancy (OR 3.50, 95% CI (1.32–9.25)), number of deliveries (OR 1.28, 95% CI (1.05–1.56)), and hypertensive disorders in pregnancy (OR 2.25, 95% CI (1.27–4.07) are significant risk factors for CP following placental abruption [22]. Moreover, heavy alcohol consumption may cause neurodevelopmental abnormalities, such as fetal alcohol syndrome and CP [73]. Smoking and hypertensive disorders also contribute to endothelial dysfunction [74,75] and uteroplacental insufficiency [76], potentially leading to higher rates of maternal, fetal, and infant mortality, and severe morbidity [77].

Our meta-analysis did not show a difference in the odds of IVH. It is possible that this meta-analysis did not have sufficient power to detect statistical differences. Spinillo et al. [45] showed significantly higher risk of grades III and IV intraventricular hemorrhage in comparison with control subjects (OR 3.5; 95% CI, 1.01 to 12.2), and even after adjusting for confounding factors, the difference remained statistically significant (OR 4.8; 95% CI 1.2 to 19.3). However, two additional studies [30,50] that assessed severe IVH did not show a significant difference between the abruption and absence of abruption groups.

Despite these findings, a previous study demonstrated a simultaneously high prevalence of low Apgar scores, prematurity, acidosis, and perinatal asphyxia, among IVH infants [45]. Moreover, among preterm infants born to mothers with placental abruption, there could have been differences in prenatal care received, differences in gestational age at birth, or complex vascular placental diseases, which could heighten the risk of IVH [78]. IVH can result from the interruption of maternal–fetal exchange, which impairs blood flow [79,80]. The presence of coagulation abnormalities associated with fetal asphyxia could also elevate the risk of IVH [79].

We did not note any statistical difference in the odds of PVL between women with placental abruption. Two of our included studies reported no significant difference in PVL rates between the cases and controls [45,50]. However, histopathological lesions show disturbance of uteroplacental circulation, including placental abruption, and are more common in infants with PVL [81]. Only one study [46] documented HIE and showed that the development quotient of children in the study in their Gesell Developmental Scale was significantly lower in the placental abruption group than in the no-abruption group, indicative of greater neurological impairment.

In contrast, a previous study reported higher occurrence of severe asphyxia in infants with PVL born to mothers with placental abruption, explaining how episodes of prolonged hypoxia result in severe metabolic acidosis, followed by higher rates of motor and cognitive deficits in surviving infants [82]. Gonen et al. found that severe neonatal morbidity (≥ 1 severe neonatal complications: seizures, IVH, HIE, PVL, blood transfusion, NEC, or death) was independently associated with early abruption (aOR = 5.3, 95% CI = 3.9–7.6), placental maternal vascular malperfusion (MVM) (aOR = 1.5, 95% CI = 1.2–1.9), and placental maternal inflammatory response (MIR) lesions (aOR = 1.9, 95% CI = 1.4–2.3) [83]. Early occurrence of abruption leads to higher rates of MVM, MIR and placental hemorrhage. In a setting of preeclampsia, other studies have suggested that MVM and MIR lesions are independently associated with lower gestational age at delivery and adverse neonatal outcomes, such as small-for-gestational age, cerebral morbidity or death [84,85]. Moreover, Sehgal et al. found that maternal vascular malperfusion, accelerated villous maturation, and fetal vascular malperfusion were features that were significantly more common in preterm fetal growth restriction (FGR) placentae than in preterm appropriately grown infants [86]. Preterm births have in turn been associated with long-term adverse neonatal outcomes, such as autism, HIE, and other neurodevelopmental disabilities [87,88,89,90].

### 4.3. Limitations

All studies in the meta-analysis were retrospective in nature and primarily relied on clinical diagnosis of placental abruption. Hence, exploration of additional variables or potential confounders, is not possible. In addition, there is no true gold standard for diagnosing placental abruption. As such, abruption was diagnosed differently across the studies, including referencing clinical diagnoses, ultrasonography, and histopathology findings. Hence, variation in the field makes it challenging to compare findings across studies and to make definitive clinical recommendations. In general, the prevalence of placental abruption is low [91], and the incidence of CP and neurological complication appears even lower. The attributable fraction for the abruption–neurological outcome association is likely very small with minimal population impact, given that abruption is unpreventable, and no existing interventions can treat it [38].

There were more preterm births in women with placental abruption than without abruption; therefore, adjusting for lower gestational age (and thus prematurity) would have been ideal to accurately examine the impact of placental abruption on cerebral palsy. Only two of the eight studies had any follow-up on infants from pregnancies complicated by placental abruption. Therefore, these studies had a poor score on the Newcastle–Ottawa scale. The absence of follow-up is also a limitation since there is evidence of increased risk of neurodevelopmental abnormalities in childhood due to prenatal hypoxia during pregnancy [92]. Hence, it is likely that these studies may have underreported cerebral palsy or other neurodevelopmental outcomes. There was overlap of data (double counting) in two articles authored by the same research group [44,45]; however, we did not include both studies in the meta-analysis, as that would have skewed numbers and led to biased results. Instead, we only included one article [45], which had a robust number of controls, and whose cases of abruption were not divided according to clinical grade of IVH. Since our sample size was small, we were unable to perform subgroup analysis to examine confounders of interest such as maternal alcohol consumption, smoking, prematurity, and other parameters. Most of the data reported in our included studies were too variable, as indicated by high I^2^ in the meta-analysis. Therefore, more consistent measures to comprehensively analyze neurodevelopmental outcomes will be needed in future placental abruption studies.

### 4.4. Conclusions and Implications

Our meta-analysis loosely suggests increased odds of cerebral palsy in infants born to pregnant women with placental abruption as compared to pregnant women without abruption. Whether this finding suggests a need for timely obstetric delivery is not presently known, given the complexity of cerebral palsy, the multiple conditions contributing to placental abruption, and the range of presentation of placental abruption at different gestational ages. Moreover, most studies do not provide an adequate follow-up period for cerebral palsy diagnosis. Our systematic review shares a promising result in that it is the first of its kind to investigate the role of placental abruption and neurodevelopmental outcomes, in a systematic manner. This is an area where there is a paucity of evidence that needs to be rectified to ensure better quality of evidence-based care for abruption patients. Future studies should further evaluate differences in neonatal neurodevelopmental outcomes in the settings of acute and chronic abruption.

## Figures and Tables

**Figure 1 jcm-12-00205-f001:**
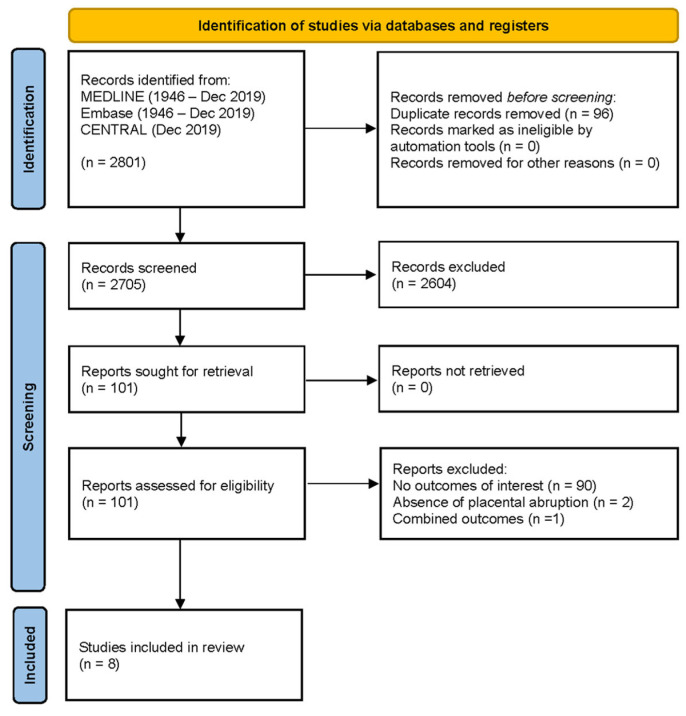
PRISMA flow diagram.

**Figure 2 jcm-12-00205-f002:**
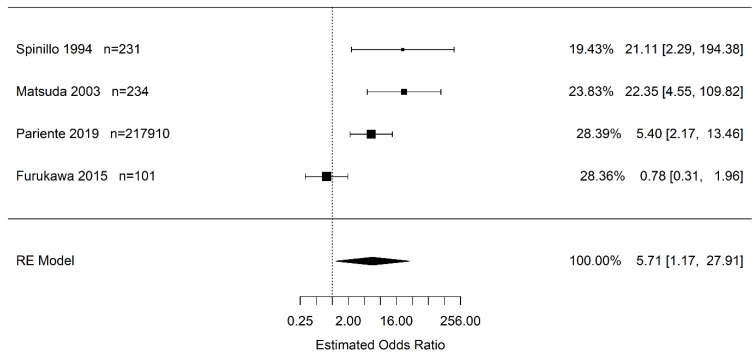
Estimate combined odds and hazard ratio of placental abruption (vs. “controls”) for cerebral palsy calculated using random effects (RE) model [24,30,45,47].

**Figure 3 jcm-12-00205-f003:**
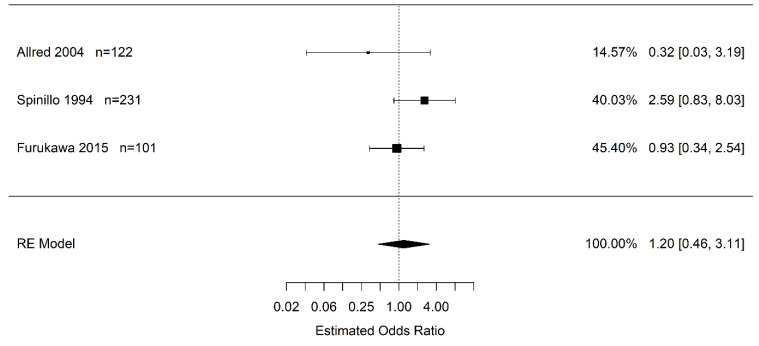
Estimate odds ratio of placental abruption (vs. “controls”) for severe intraventricular hemorrhage (grade 3, 4) calculated using random effects (RE) model [30,45,50].

**Figure 4 jcm-12-00205-f004:**
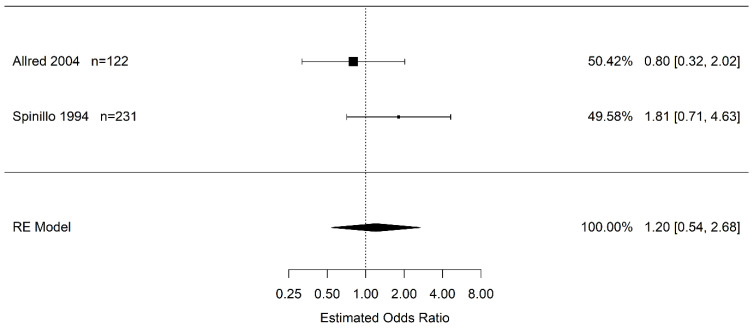
Estimate odds ratio of placental abruption (vs. “controls”) for intraventricular hemorrhage (any grade) calculated using random effects (RE) model [45,50].

**Figure 5 jcm-12-00205-f005:**
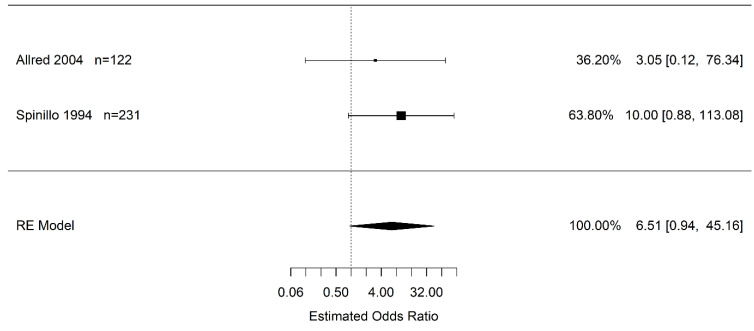
Estimated odds ratio of placental abruption (vs. “controls”) for periventricular leukomalacia calculated using random effects (RE) model [45,50].

**Table 1 jcm-12-00205-t001:** Study characteristics.

Authors, Location (Year)	Inclusion Criteria and Dates; Data Source	Sample Size (Abruptions)	Sample Size (Non-Placental Abruption Pregnancies)	Cerebral Palsy (n)	Intraventricular Hemorrhage (n)	Hypoxic Ischemic Encephalopathy (n)	Periventricular Leukomalacia (n)	Histopathological Neuronal Damage (n)	Neonatal Mortality (n)
Cohort studies									
Pariente et al., Israel (2019) [24]	All women delivering at the Soroka University Medical Center (SUMC) from 1991 to 2014; single tertiary centre	1003	216,907	8	Not reported	Not reported	Not reported	Not reported	Not reported
Becher et al., Scotland (2006) [51]	All perinatal deaths that were ≥ 24 weeks gestation at birth and ≤ 7 days at time of death from 1996 to 1999; 22 obstetric units	9	182	Not reported	Not reported	Not reported	Not reported	5	Not reported
Case control studies									
Lv et al., China (2019) [46]	All neonates with (1) gestational age ≥ 37 weeks and weight ≥ 2500 g; (2) severe asphyxia at birth indicated by 1 min Apgar score ≤ 3 or a 5 min Apgar score ≤ 5; (3) umbilical artery blood pH ≤ 7 at birth; (4) abnormal neurological signs in the first 24 h of life from 2013 to 2017; single site	18	35	Not reported	Not reported	Case: Moderate HIE: 7Severe HIE: 11 Control: Moderate HIE: 0 Severe HIE: 0	Not reported	Not reported	Not reported
Furukawa et al., Japan (2015) [30]	Infants born at 22 to around 26 weeks of gestation to women with placental abruption from 2000 to 2010; controls were infants born without abruption within 1 week of the non-control gestational age and birth weight of 50 g; single site	32	69	Case: 9 Control: 23	Case: Grades III–IV IVH: 7 Control: Grades III–IV IVH: 16	Not reported	Not reported	Not reported	Case: 6 Control: 8
Allred et al., USA (2004) [50]	Live-born infants between 23 and 32 weeks gestation born to women with placental abruption from 1995 to 1999; controls were infants who did not suffer abruption within 100 g of weight and 1 week of gestation; single site	61	61	Not reported	Case: Unspecified IVH Grade 1 placental abruption: 6 Grade 2 placental abruption: 4 Grade III–IV IVH Grade 1 placental abruption: 1 Grade 2 placental abruption: 0 Control: Unspecified IVH: 0 Grades III-IV IVH: 3	Not reported	Case: Grade 2 placental abruption: 1 Control: 0	Not reported	Not reported
Matsuda et al., Italy (2003) [47]	Live singleton births between 26 and 36 weeks of gestation between 1992 and 1999; single site	42	120	Case: 8 Control: 1	Not reported	Not reported	Not reported	Not reported	Case: 3 Control: 0
Spinillo et al., Italy (1994) [45]	Singleton, liveborn low birthweight infants from 1983 to 1989; single site (database)	40	154	Not reported	Case: Grades I–II: 2 Grades III–IV: 5 Control: Grades I–II: 10 Grades III–IV: 6	Not reported	Case: 2 Control: 1	Not reported	Case: 4 Control: 19
Spinillo et al., Italy (1993) [44]	Singleton, liveborn low birthweight infants from 1983 to 1989; single site (database)	40	80	Case: Group 1 placental abruption: 1 Group 2 placental abruption: 1 Group 3 placental abruption: 2 Control: 0	Case: Grade I-II IVH Group 1 placental abruption: 1 Group 2 placental abruption: 1 Group 3 placental abruption: 1 Grade III–IV IVH:Group 2 placental abruption: 1 Group 3 placental abruption: 3 Control: Grade I–II IVH: 3 Grade III–IV IVH: 1	Not reported	Case: Group 2 placental abruption: 1 Group 3 placental abruption: 1 Control: 0	Not reported	Case: Group 1 placental abruption: 1 Group 2 placental abruption: 1 Group 3 placental abruption: 2 Control: 6

**Table 2 jcm-12-00205-t002:** Maternal characteristics of the study population.

Authors, Location (Year)	Mean Maternal Age in Years (SD)	Comorbidities (%)	Parity (%)	Caesarean Delivery (%)
**Cohort studies**				
Pariente et al., Israel (2019) [24]	Placental abruption: 29.6 (6.2) No placental abruption: 28.2 (5.7)	Placental abruption: Chronic hypertension—3.3 Gestational hypertension—12.3 Pre-eclampsia—10.5 Pre-gestational diabetes mellitus—7.5 Gestational diabetes mellitus—7.5 Smoking—1.4 No placental abruption:Chronic hypertension—1.4 Gestational hypertension—5.2 Pre-eclampsia—4.1 Pre-gestational diabetes mellitus—5.4 Gestational diabetes mellitus—5.5 Smoking—1.1	Placental abruption: One—23.5 Two—19.2 Three or more—57.2 No placental abruption: One—24.8 Two—22.8 Three or more—52.4	Placental abruption: 75.8 No placental abruption: 13.8
Becher et al., Scotland (2006) [51]	28.8 (6.6)	Complicated pregnancy: 64	Multiple gestation: 7	N/A
**Case control studies**				
Lv et al., China (2019) [46]	Placental abruption: 25.7 (20.0, 37.0) (median, min–max) Control: 26.1 (18.0, 39.0) (median, min–max)	Not reported	Not reported	Placental abruption: 38.9 Control: 42.9
Furukawa et al., Japan (2015) [30]	Placental abruption: 29.3 (5.8) Control: 29.9 (4.9)	Placental abruption: Hypertension—1 Control: Hypertension—1	Placental abruption: Primipara: 41 Control: Primipara: 32	Placental abruption: 56 Control: 67
Allred et al., USA (2004) [50]	Not reported	Not reported	Not reported	Grade 1 placental abruption: 53 Grade 2 placental abruption: 96 Control group 1: 42 Control group 2: 78
Matsuda et al., Italy (2003) [47]	Placental abruption: 31.5 (5.1) Control: 28.7 (4.4)	Not reported	Placental abruption: Multipara: 42.9 Control: Multipara: 30	Placental abruption: 90.4 Control: 47.5
Spinillo et al., Italy (1994) [45]	Placental abruption: 26.9 (3.8) Control: 28.3 (4.9)	Placental abruption:Smoking: 32.5 Control: Smoking: 30	Placental abruption: 1 (20) ≥ 2 (27.5) Control: 1 (28.6) ≥ 2 (22.1)	Placental abruption: Elective—17.5 Emergency—52.5 Control: Elective—31.2 Emergency—13.6
Spinillo et al., Italy (1993) [44]	Placental abruption: 26.9 (3.8) Control: 28.5 (4.5)	Placental abruption: Smoking—32.5 Control: Smoking—30	Placental abruption: One—20 ≥ Two: 27.5 Control: One—28.6 ≥ Two—22.1	Placental abruption: Elective—17.5 Emergency—52.5 Control: Elective—45 Emergency—10

**Table 3 jcm-12-00205-t003:** Neonatal characteristics of the study population.

Authors, Location (Year)	Mean Gestational Age in Weeks (SD)	Mean Birth Weight in g, (SD)	Preterm Delivery (%)	Apgar Scores (%)	Cord Gas pH (%)	Male Sex (%)	Caesarean Delivery (%)
**Cohort studies**							
Pariente et al., Israel (2019) [24]	Placental abruption: 36.5 (3.5) Control: 39.1 (1.7)	Placental abruption: 2630 (76) Control: 3224 (48)	Placental abruption: 42.9 Control: 6.2	Placental abruption: 1 min Apgar < 4: 11.2 5 min Apgar < 4: 0.89 Control: 1 min Apgar < 4: 0.7 5 min Apgar < 4: 0.08	Not reported	Placental abruption: 53.2 Control: 50.9	Placental abruption: 75.8 Control: 13.8
Becher et al., Scotland (2006) [51]	33.4 (5.3)	Not reported	Growth restricted preterm: 23	Not reported	Not reported	48	Not reported
**Case control studies**							
Lv et al., China (2019) [46]	Placental abruption: 38.6 (37.0, 41.0) (median, min–max) Control: 39.1 (37.0, 41.0) (median, min–max)	Placental abruption: 3290 (2550, 4100) (median, min–max) Control: 3325 (2500, 4600) (median, min–max)	Not reported	Placental abruption: 1 min Apgar score: 1.91 (1.23, 2.67) (median, min–max) 5 min Apgar score: 3.25 (1.76, 4.51) Control: 1 min Apgar score: 8.48 (7.63, 9.52) (median, min–max) 5 min Apgar score: 9.52 (8.35, 9.61)	Not reported	Placental abruption: 50 Control: 60	Placental abruption: 38.9 Control: 42.9
Furukawa et al., Japan (2015) [30]	Placental abruption: 24.2 (1.3) Control: 24.2 (1.2)	Placental abruption: 649 (143) Control: 643 (125)	Not reported	Not reported	Cord pH < 7.1 Placental abruption: 0 Control: 3	Placental abruption: 56 Control: 49	Placental abruption: 56 Control: 67
Allred et al., USA (2004) [50]	Grade 1 placental abruption: 29.4 (2.6) Grade 2 placental abruption: 27.5 (2.9) Control group 1:29.4 (2.6) Control group 2: 27.6 (2.8)	Grade 1 placental abruption: 1358 (402) Grade 2 placental abruption: 1089 (414) Control group 1: 1369 (388) Control group 2: 1124 (428)	Not reported	Grade 1 placental abruption: Apgar score ≤ 3 at 1 min: 8 Apgar score ≤ 5 at 5 min: 0 Grade 2 placental abruption: Apgar score ≤ 3 at 1 min: 35 Apgar score ≤ 5 at 5 min: 17 Control group 1: Apgar score ≤ 3 at 1 min: 5 Apgar score ≤ 5 at 5 min: 0 Control group 2: Apgar score ≤ 3 at 1 min: 13 Apgar score ≤ 5 at 5 min: 0	Cord pH < 7.0: Grade 1 placental abruption: 0 Grade 2 placental abruption: 22 Control: Group 1: 0 Group 2: 0	Not reported	Grade 1 placental abruption: 53 Grade 2 placental abruption: 96 Control group 1: 42 Control group 2: 78
Matsuda et al., Italy (2003) [47]	Placental Abruption: 31.2 (3.4) Control: 30.5 (3.8)	Placental abruption: 1670 (533) Control: 2010 (583)	Not reported	Placental abruption: Apgar score at 1 min (<7): 46 Apgar score at 5 min (<7): 26.2 Control: Apgar score at 1 min (<7): 19.2 Apgar score at 5 min (<7): 0.8	Fetal acidemia (<7.0): Placental abruption: 33.3 Control: 0.8	Not reported	Placental abruption: 90.4 Control: 47.5
Spinillo et al., Italy (1994) [45]	Placental abruption 33.4 (3.4) Control: 32.7 (4.0)	Placental abruption: 1800 (458) Control: 32.7 (4.0)	Not reported	Not reported	Acidemia (pH < 7.2 in the first 12 h): Placental abruption: 30 Control: 16.2	Not reported	Placental abruption: Elective caesarean—17.5 Emergency caesarean—52.5 Control: Elective caesarean—31.2 Emergency caesarean—13.6
Spinillo et al., Italy (1993) [44]	Grade 1 placental abruption: 33.1 (3.8) Grade 2 placental abruption: 32.9 (43.2) Grade 3 placental abruption: 33.9 (3.3) Control: 33.4 (3.3)	Grade 1 placental abruption: 1801 (476) Grade 2 placental abruption: 1760 (506) Grade 3 placental abruption: 1855 (430) Control: 1811 (553)	Not reported	Grade 1 placental abruption: Apgar score at 1 min 7.2: 2.0 Apgar score at 5 min 8.4: 1.6 Grade 2 placental abruption: Apgar score at 1 min: 6.6 (2.5) Apgar score at 5 min: 7.2 (2.2) Grade 3 placental abruption: Apgar score at 1 min: 5.2 (3.0) Apgar score at 5 min: 6.6 (2.4) Control: Apgar score at 1 min: 7.1 (2.6) Apgar score at 5 min: 8.2 (1.8)	Acidemia (pH < 7.2 in the first 12 h: Grade 1 Placental abruption: 16.7 Grade 2 placental abruption: 16.7 Grade 3 placental abruption: 50.0 Control: 20.0	Grade 1 placental abruption: 25.0 Grade 2 placental abruption: 50.0 Grade 3 placental abruption: 43.7 Control: 58.8	Placental abruption: Elective caesarean—17.5 Emergency caesarean—52.5 Controls: Elective caesarean—45 Emergency caesarean—10

**Table 4 jcm-12-00205-t004:** Assessment of study quality using the Ottawa–Newcastle scale. ★ Star denotes that the specific criteria are acceptable.

Study	Selection				Comparability	Outcome			Quality SCORE
	Representativeness of Cases	Selection of Controls	Ascertainment of Abruption	Outcome Assessed Prior to Study	Comparability of Study Groups	Assessment of Outcome	Long Follow-Up	Follow-Up Adequacy	
Allred et al., (2004) [50]	Participants were from one hospital ★	Derived from same population ★	Hospital records ★	No	No mention of matching or adjustment	Record linkage ★	No	No	Poor
Matsuda et al., (2003) [47]	Participants were from one hospital ★	Derived from same population ★	Hospital records including imaging and pathology reports ★	No	Matched/adjusted for confounding factors ★	Record linkage ★	No	No	Fair
Spinillo et al., (1994) [45]	Participants were from one hospital ★	Derived from same population ★	Hospital records including imaging and pathology reports ★	No	Matched/adjusted for confounding factors ★	Record linkage ★	Yes, at least 24 months ★	Follow-up rate less than 95% and no description of those lost ★	Good
Pariente et al., (2019) [24]	Participants were from one hospital ★	Derived from same population ★	Hospital records ★	No	Matched/adjusted for confounding factors ★	Record linkage ★	Yes, at least 24 months ★	Complete follow-up, that is, all subjects accounted for ★	Good
Furukawa et al., (2015) [30]	Participants were from one hospital ★	Derived from same population ★	Hospital records ★	No	No mention of matching or adjustment	Independent blind assessment ★	No	No	Poor
Becher et al., (2005) [51]	Participants were from several hospitals ★	Derived from same population ★	Interview of caregivers—not blinded	No	No mention of matching or adjustment	Independent blind assessment ★	No	No	Poor
Lv et al., (2018) [46]	Participants were from one hospital ★	Derived from same population ★	Hospital records including imaging ★	No	No mention of matching or adjustment	Record linkage ★	No	No	Poor
Spinillo et al., (1993) [44]	Participants were from one hospital ★	Derived from same population ★	Hospital records including imaging and pathology reports ★	No	Matched/adjusted for confounding factors ★	Record linkage ★	Yes, at least 24 months ★	Follow-up rate less than 95% and no description of those lost ★	Good

## Data Availability

The datasets generated and/or analyzed during the current study are available from the corresponding author upon reasonable request.

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
