# Peer review of "The Association of Placental Abruption and Pediatric Neurological Outcome: A Systematic Review and Meta-Analysis"

_jcm, 2022, doi:10.3390/jcm12010205_

Round 1
Reviewer 1 Report
Dear Author (s)
1. The number of studies in the meta-analysis is very low. Therefore, the results are not valid.
2. There are a lot of grammatical errors such as: "had complete" should be "had completed", etc.
3. Some columns in Table 1 are not necessary due to a lack of information for most studies.
4. Please add the full name of each abbreviation for each table or figure.
Author Response
We would like to thank the reviewer for her/his comments. Please find our responses to the concerns (in red font). The revised manuscript with track changes turned on is attached.
1. The number of studies in the meta-analysis is very low. Therefore, the results are not valid.
We completely agree. We have therefore mentioned several times in the manuscript, that this is a major limitation in the study. We have now made it more clear that the sample size is a limitation and this major gap should be addressed in future primary studies.
We have added additional content in the conclusion, so this paucity of evidence is highlighted.
2. There are a lot of grammatical errors such as: "had complete" should be "had completed", etc.
Thank you for noting such errors. Several changes have been made in the manuscript to resolve such errors.
3. Some columns in Table 1 are not necessary due to a lack of information for most studies.
While most do not, some of the studies (or at times just 1 study) have details pertaining to the column. For the sake of providing complete details, we included all these columns.
4. Please add the full name of each abbreviation for each table or figure.
We have removed abbreviations from tables and figure captions.
Reviewer 2 Report
he reviewed meta-analysis is well prepared in accordance with the recommended guidelines. From my point of view, appropriate methods were used. The validity of the results is affected by the quality of the included studies, the heterogeneity of the studies, and the number of included subjects. As for the primary outcome, I consider the sample size appropriate, and the result could be regarded as valid, although the confidence interval is rather high. In terms of secondary outcomes, small sample sizes do not allow for valid results. Of course, the authors cannot be blamed since they were not aware of this before conducting a literary search. The authors clearly stated the limitations of the results.
The results are clearly presented. The manuscript requires professional English language editing.
Minor comments:
Hypercarbia – consider changing to hypercapnia
Do not write „:“ after OR
stable, can. – omit „ , “
hypoxic ischemic or hypoxic-ischemic - need to be unified and reported correctly
palsy is however inconclusive – correct to palsy is, however, inconclusive
add comma after vs
Author Response
We would like to thank the reviewer for their comments. Please find our reply to the concerns below (in red font). Besides these, several other revisions were made for consistency in the manuscript. The revised manuscript with track changes turned on is also attached.
1. Hypercarbia – consider changing to hypercapnia
Thank you for this suggestion. We have revised it.
2. Do not write „:“ after OR
We have removed ':' after OR
3. stable, can. – omit „ , “
This sentence has been revised.
4. hypoxic ischemic or hypoxic-ischemic - need to be unified and reported correctly
We have retained 'hypoxic ischemia' and made it consistent.
5. palsy is however inconclusive – correct to palsy is, however, inconclusive
We have made this change.
6. add comma after vs
We decided to go with vs instead of versus and have included 'vs.' consistently throughout the manuscript.